# Pro-Calcifying Role of Enzymatically Modified LDL (eLDL) in Aortic Valve Sclerosis via Induction of IL-6 and IL-33

**DOI:** 10.3390/biom13071091

**Published:** 2023-07-07

**Authors:** Annemarie Witz, Denise Effertz, Nora Goebel, Matthias Schwab, Ulrich F. W. Franke, Michael Torzewski

**Affiliations:** 1Dr. Margarete Fischer-Bosch Institute of Clinical Pharmacology, 70376 Stuttgart, Germany; 2Department of Cardiovascular Surgery, Robert-Bosch-Hospital, 70376 Stuttgart, Germany; 3Department of Clinical Pharmacology, University of Tuebingen, 72076 Tuebingen, Germany; 4Department of Biochemistry and Pharmacy, University of Tuebingen, 72076 Tuebingen, Germany; 5Department of Laboratory Medicine and Hospital Hygiene, Robert-Bosch-Hospital, 70376 Stuttgart, Germany

**Keywords:** enzymatically modified LDL (eLDL), calcification, aortic valve stenosis, VICs/myofibroblasts, p38 MAPK, IL-6, IL-33

## Abstract

One of the contributors to atherogenesis is enzymatically modified LDL (eLDL). eLDL was detected in all stages of aortic valve sclerosis and was demonstrated to trigger the activation of p38 mitogen-activated protein kinase (p38 MAPK), which has been identified as a pro-inflammatory protein in atherosclerosis. In this study, we investigated the influence of eLDL on IL-6 and IL-33 induction, and also the impact of eLDL on calcification in aortic valve stenosis (AS). eLDL upregulated phosphate-induced calcification in valvular interstitial cells (VICs)/myofibroblasts isolated from diseased aortic valves, as demonstrated by alizarin red staining. Functional studies demonstrated activation of p38 MAPK as well as an altered gene expression of osteogenic genes known to be involved in vascular calcification. In parallel with the activation of p38 MAPK, eLDL also induced upregulation of the cytokines IL-6 and IL-33. The results suggest a pro-calcifying role of eLDL in AS via induction of IL-6 and IL-33.

## 1. Introduction

Aortic valve stenosis (AS) constitutes an immense clinical and economic burden, which is expected to increase with the aging population in the near future [1,2]. Despite its high prevalence, the underlying pathophysiological pathways from inflammation to calcification, finally leading to severe stenosis, remain incompletely understood [3]. AS is characterized by fibrocalcific remodeling of the valve leaflets resulting in progressive narrowing of the aortic valve opening, restriction of blood flow, and consecutive pressure load on the left ventricle. The consequences are far-reaching, with symptoms ranging from angina pectoris or syncope to sudden cardiac death [4,5]. To date, no medical treatment has proven to be effective in halting or reducing the disease’s progression [6,7]. Therefore, conventional surgical or transcatheter aortic valve replacement (AVR) remains the only available treatment option for severe AS [8,9]. However, the risks of the surgical procedures prevent elderly patients and patients with severe comorbidities from undergoing optimal treatment. It is, therefore, important to investigate the underlying mechanisms of AS, which could provide new insights for the development of pharmacological interventions.

AS has long been considered a passive degenerative disease in which “wear and tear” over the years leads to a gradual accumulation of calcium in the valve leaflets. It is now clear that AS is rather the result of an actively regulated and complex cellular process [4,5,10]. The pathophysiology of AS is considered to occur in two phases: the initiation phase, similar to atherosclerosis, is characterized by lipid infiltration and deposition and accompanied by inflammation, and the propagation phase, in which myofibroblastic/osteoblastic differentiation of valvular interstitial cells (VICs), fibrosis, and calcification are responsible for disease progression [11,12,13].

The presence of common risk factors and genetic dispositions of atherosclerosis and AS highlights the existence of shared mechanisms for disease initiation. In both diseases, endothelial injury due to increased mechanical and shear stress, followed by insudation and accumulation of lipoproteins in the intima and fibrosa, respectively, are thought to be the initiating events [14]. These lipoproteins can undergo several modifications and become cytotoxic, which are then capable of stimulating inflammatory activity. Amongst others, in particular the widespread oxidation hypothesis, we propagated the less common eLDL hypothesis, which proposes that modification of LDL occurs through the action of ubiquitous hydrolytic enzymes (enzymatically modified LDL or eLDL) rather than oxidation [15,16]. Initially, enzymatic modification of LDL in vitro was performed by sequential treatment with trypsin, cholesterol esterase, and neuraminidase [17], although the former was later on replaced by several other proteases [18]. Unlike the respective shortcomings of the oxidation hypothesis, our work indicated that eLDL is present already in early atherosclerotic lesions [19] and activates the complement via CRP-dependent and -independent pathways distinguished between atherosclerotic lesion initiation with reversion or lesion initiation with progression, respectively [15,20]. Moreover, in our previous work, eLDL was detected in all stages of aortic valve sclerosis, thus, providing evidence that modified lipoproteins are linked to the pathogenesis of AS, just as in atherosclerosis [21]. In the context of chronic inflammation, the p38 mitogen-activated protein kinase (p38 MAPK) signalling pathway has gained attention from researchers in the fields of both atherosclerosis and aortic valve sclerosis. Due to their presence in atherosclerosis and aortic valve sclerosis, native LDL and its modification product eLDL may represent important triggers for p38 MAPK signalling [14]. Previous studies have shown that incubation of vascular endothelial cells with either native LDL [22] or eLDL [23] results in p38 MAPK phosphorylation. Furthermore, eLDL was demonstrated to induce the phosphorylation of p38 MAPK in human monocytes and macrophages associated with atherosclerotic plaques. Human monocytes and macrophages take up eLDL, leading to subsequent activation of the p38 MAPK pathway [24]. p38 MAPK activation in macrophages has been shown to induce the expression of pro-inflammatory cytokines in response to modified LDL [25,26].

Regarding the pro-inflammatory cytokines IL-6 and IL-33, there is initial evidence of an association with calcific AS [27,28]. However, with regard to IL-33, the data seem to be contradictory, especially when taking a comparative look at atherogenesis. On the one hand, a protective effect of IL-33 has been observed in animal models [29,30]. On the other hand, an inverse association of IL-33 serum levels and the severity of CHD (coronary heart disease) was found [31]. Likewise, the cytokines’ effect in the pathogenesis of AS remains undiscovered. Accordingly, the current study investigates the interaction between eLDL and VICs with particular attention to the p38 MAPK pathway and the pro-inflammatory cytokines IL-6 and IL-33.

## 2. Materials and Methods

### 2.1. Human Aortic Valvular Tissue

Human aortic valves were obtained from 36 patients undergoing aortic valve replacement surgery. Table 1 shows the detailed characteristic of the patients. Data are expressed as median (minimum difference, maximum difference, interquartile range) or (percentage) number of subjects. All patients gave written informed consent, and the study was approved by the local Ethics Committee of the Medical Faculty and the University Hospital of Tuebingen, Germany. Immediately after surgical removal, the valves were immersed in PBS and stored at 4 °C until transport to the laboratory.

### 2.2. Histological and Immunhohistochemical Analysis

Sections of formalin-fixed non-rheumatic stenotic aortic valves with varying degrees of macroscopic disease were taken vertically through Grade 1 to 4 lesions according to Warren and Young [32]. For histochemistry, paraffin-embedded specimens were stained with Elastica van Gieson (EvG) and Masson–Goldner to illustrate the layered architectural pattern. Immunohistochemistry was performed using the Dako Real^TM^ Envision^TM^ detection system, rabbit/mouse kit (Dako, Glostrup, Denmark). Serial 3 µm thick sections of paraffin-embedded aortic valve leaflets were deparaffinized and treated with Dako Real^TM^ peroxidase-blocking solution for 10 min to block endogenous peroxidase activity. After blocking, slides were incubated with the primary antibodies listed in Table 2. Antigen retrieval of IL-6 (1:100 in Dako Real™ antibody diluent; Dako) and IL-33 (1:100 in Dako Real™ antibody diluent; Dako) was achieved by heating the sections in pre-warmed target retrieval solution pH 6 and pH 9 (Dako) in a steamer for 30 min. Application of the primary antibody was followed directly by the secondary antibody for 30 min. The reaction products were detected by immersing the slides in diaminobenzidine tetrachloride (DAB) for 10 min, resulting in a brown reaction product. Finally, the slides were counterstained with hemalaun (Papanicolaou’s solution 1a Harris’ hematoxylin; Merck, Darmstadt, Germany) and mounted using Neo-Mount (Merck).

### 2.3. VIC/Myofibroblast Isolation and Culture

VICs/myofibroblasts were isolated from stenotic aortic valves obtained from valve replacement surgery. Isolation was performed immediately after surgical removal of the human aortic valves via sequential collagenase digestions, using a modified method from Schlotter et al. [33]. Briefly, both sides of the leaflet were scratched with a razor blade to remove the endothelial cells. Tissue pieces were digested using a sterile filtered collagenase solution, containing cell rinse buffer (120 mM NaCl, 15.6 mM glucose, 2.5 mM MgCl_2_ × 6 H_2_O, 5.4 mM KCl, 1 mM NaH_2_PO_4_, 20 mM HEPES (pH 7.2)), type I collagenase (125 U/mg; Sigma-Aldrich) and protease (0.25 mg/mL; Sigma-Aldrich) for 90 min at 37 °C, 5% CO_2_, with gentle mixing every 20 min. The digested valve was then passed through a 0.2 µm filter, and the flowthrough was centrifuged at 1400 rpm for 5 min. VICs/myofibroblasts were collected by centrifugation, resuspended in 5 mL Dulbecco’s modified Eagle’s medium (DMEM; Fisher Scientific by Thermo Fisher Scientific) supplemented with 10% fetal calf serum (FCS; Thermo Fisher Scientific), 1% penicillin–streptomycin (pen/strep; Fisher Scientific) and 1% fungizone (Amphotericin B; Gibco by Thermo Fisher Scientific) and plated in 25 cm^2^ culture flasks. After three days, the medium was changed to DMEM supplemented with 10% FCS and 1% pen/strep and the isolated VICs/myofibroblasts were grown to confluence before passage. Cells were then trypsinized (trypsin-EDTA; Sigma-Aldrich), counted, and plated for each experiment. Cells between passages 2 and 4 were used for all experiments.

### 2.4. Treatment of VICs/Myofibroblasts

Experiments were performed on early passage cells (2–4) from several different patients (indicated by n number). VICs/myofibroblasts were plated on different cell culture plates and serum-starved by culturing in fetal calf serum-free DMEM medium for 24 h before experimental use. Cells were cultured in the absence or presence of inorganic phosphate (1 mM) and treated with varying concentrations of eLDL (2.5–40 μg/mL), IL-6 (100 ng/mL) or IL-33 (100 ng/mL). The doses of eLDL, IL-6, and IL-33 used in this study were determined based on various preliminary experiments and previously published in vitro experiments (cf. Chellan et al. (2.5 and 5 μg/mL eLDL), Zhu and Carver and Wang et al. (100 ng/mL IL-33)) [34,35,36]. DMEM supplemented with inorganic phosphate (NaH_2_PO_4_) was used to promote calcification in VICs/myofibroblasts. Cells were incubated with the respective reagents and harvested after the indicated time points. DMEM served as a control treatment.

### 2.5. Lipoprotein Isolation and Modifications

Human LDL was isolated via preparative ultracentrifugation in KBr gradients followed by extensive dialysis against buffer containing 150 mM NaCl, 5 mM Tris-Base, 2.7 mM EDTA × 2 H_2_O, pH 7.3–7.4, and filter sterilization. Enzymatically modified LDL (eLDL) was prepared as previously described with some modifications [37]. For enzymatic modification, 1 mL samples of native LDL containing 3 mg/mL cholesterol were digested with plasmin (0.1 U/mL; Merck) and incubated on a shaker overnight at 37 °C. Then, 40 μL 25× CPI (complete proteinase inhibitor; Roche, Basel, Switzerland) and CE (cholesterol esterase, 35 U/mg; Sigma-Aldrich) were added to cleave the cholesterol inside the LDL particle and incubated for another 24 h at 37 °C in a shaker until the eLDL preparation appeared cloudy. To monitor the reactions, the turbidity was measured at 595 nm [18]. Protein content was determined by the Bradford method before aliquots of eLDL were stored at 4 °C. eLDL was tested for endotoxins using the Kit Endonext (Biomerieux, Nuertingen, Germany), and only negative samples were used for experiments.

### 2.6. Calcification Assay

VICs/myofibroblasts were cultured in pro-calcifying medium (termed PM, Chellan et al. [34]). Cells between passages 2 and 4 were used from different batches of VICs/myofibroblasts. Cells were grown to confluency in 24-well dishes using DMEM supplemented with 10% FCS; 1% pen/strep at 37 °C, and 5% CO_2_. After confluence, VICs/myofibroblasts were incubated overnight in FCS-free DMEM. The next day, the medium was replaced with PM containing DMEM with 0.1% FCS, 1% pen/strep, and 0.5–1 mM of inorganic phosphate (NaH_2_PO_4_). Regular advanced DMEM contains 1 mM phosphate. As indicated, eLDL (2.5–5 µg/mL) or IL-6 (100 ng/mL) were added, and cells were cultured for 3 to 7 days. PM was replaced every 3 days. After PM removal, VICs/myofibroblasts monolayers were rinsed with PBS and fixed in 4% formaldehyde for 10 min at room temperature. Calcium deposits were visualized via staining with alizarin red solution. Fixed cells were incubated in an aqueous solution of 1% alizarin red solution (pH 4.1–4.4; Roth, Karlsruhe, Germany) for 10 min and washed three times with distilled water to remove unbound stain. Quantification of the alizarin–calcium complexes was carried out according to a method described by Prosdocimo et al. [38]. Briefly, the deposited alizarin–Ca^2+^ complexes were extracted by adding a 100 mM cetylpyridinium chloride solution (CPC). The optical density (OD) of the samples was measured at 570 nm and normalized to total cellular protein (Bradford [39]).

### 2.7. RNA Isolation and Real-Time Quantitative PCR

Total RNA from VICs/myofibroblasts of human aortic valves was isolated using a glass fiber filter-based method (mirVana miRNA Isolation Kit; Invitrogen by Thermo Fisher Scientific) according to the manufacturer’s instructions. Cells were loaded with eLDL (5–40 µg/mL) or IL-33 (100 ng/mL) in 12-well plates and incubated for the indicated times (1/2 h, 1 h, 6 h, 8 h, 12 h, 24 h, 48 h, 6 days, and 7 days). For the p38 MAPK inhibition assay, VICs/myofibroblasts were pre-treated with skepinone-L (0.1 μM) or SB203580 (20 μM) and further incubated with 40 μg/mL eLDL. Untreated cells were used as control. Extraction of the RNA was performed using 300 μL lysis binding solution. The RNA was eluted in 50 μL of the elution buffer pre-warmed to 95 °C. The extracted RNA was stored in an ultra-freezer at −80 °C for use in downstream analysis. The concentration of total RNA was assessed using a NanoDrop spectrophotometer (peQLab Biotechnologie GmbH, Erlangen, Germany). Total RNA from the VICs/myofibroblasts were reverse transcribed into cDNA with a high-capacity cDNA Reverse Transcription Kit (Thermo Fisher Scientific). Relative mRNA expression of target genes was measured by quantitative polymerase chain reaction (qPCR) and normalized to housekeeping gene glyceraldehyde 3-phosphate dehydrogenase (GAPDH) according to the 2^−ddCt^ method. qPCR was performed using the 7900 HT Fast Real-Time PCR system with TaqMan Universal Master Mix II (Thermo Fisher Scientific). The TaqMan Gene Expression Assays used for qPCR amplification are shown in Table 3.

### 2.8. Western Blot Analyses

Protein expression was semi-quantified by Western blotting in whole cell lysis extract from VICs/myofibroblasts. Cells were plated in 6 cm cell culture dishes and incubated with eLDL (10–20 µg/mL), IL-6 (100 ng/mL) or IL-33 (100 ng/mL) for 1/2 h, 24 h, and 48 h. Cultured VICs/myofibroblasts were lysed directly on the culture plates in ice-cold cell signaling buffer containing cell lysis buffer (CLP 1×; Cell Signaling Technology), PhosSTOP phosphatase inhibitor (PI; Roche), cOmplete Mini protease inhibitor (CPI, Roche), and phenylmethylsulfonyl fluoride (PMSF; Sigma-Aldrich) protease inhibitor. Suspended cells were collected, and total cell proteins were extracted by centrifugation. Protein concentrations were determined according to Bradford [39]. Equal amounts of the protein samples (40 μg) were loaded onto 10% SDS gels, separated by electrophoresis and then transferred to a nitrocellulose membrane (Hybon ECL, Amersham™ Protran™ 0,45 µm NC; Amersham Pharmacia Biotech, Piscataway, NJ, USA) according to standard procedures. After blocking nonspecific sites, the membranes were incubated with the primary antibodies against total and phosphorylated p38 MAPK (Table 2) and vinculin or β-actin at a 1:1000 dilution overnight at 4 °C. After washing, the membranes were stained with horseradish peroxidase (HRP)-conjugated goat anti-rabbit and horse anti-mouse IgG secondary antibodies (Cell Signaling Technology, Danvers, MA, USA) for 1 h. HRP-conjugated secondary antibodies were used in conjunction with a lumino-based ECL (enhanced chemiluminescence) horseradish peroxidase substrate kit (SuperSignal West Dura Extended Duration, Substrate; Pierce Biotechnology, Rockford, IL, USA). Light emission was detected by the LAS1000 imaging system using STELLA and AIDA software (RayTest, Straubenhardt, Germany). Protein expression was analyzed by AIDA software and normalized to vinculin or β-actin.

### 2.9. MTT Assay

The MTT (3-(4,5-dimethylthiazol-2-yl)-2,5-diphenyltetrazolium bromide) viability assay was performed as described by Mosmann et al. [40] with slight modifications. MTT (Sigma) was prepared as a working solution of 10 mg/mL in phosphate-buffered saline (PBS, pH 7.2). At the end of the different treatment periods (see above), 10 µL of MTT solution was added to each well. After incubation at 37 °C for 2 h, 90 µL of lysis buffer (15% sodium dodecyl sulfate (SDS) dissolved in dimethylformamide (DMF) water 1:1, pH 4.5, adjusted with 80% acetic acid) was added to each well and the plate was incubated for 2 h at room temperature in the dark. The microtiter plate was placed on a shaker in order to dissolve the formazan crystals. Unlike dead cells, viable cells produced a dark purple formazan product. Cell viability was assessed by determining the absorbance at a wavelength of 550 nm using a 96-well microplate reader (Enspire; Perkin Elmer, Waltham, MA, USA).

### 2.10. Statistical Analyses

Data are represented as mean values ± standard deviation (SD). An unpaired two-tailed Student’s *t* test was used to determine statistical significance of differences in mean values. *p*-values < 0.05 were considered statistically significant (* *p* < 0.05, ** *p* < 0.01 and *** *p*< 0.001). Statistical analysis of the data were calculated and graphed using Excel and Prism 9 software (GraphPad software Inc., San Diego, CA, USA). The *p*-values are indicated in the individual figures. 

## 3. Results

### 3.1. Estimation of Cell Viability in VICs/Myofibroblasts

First of all, the effect of eLDL on cell viability was measured by MTT assay. VICs/myofibroblasts were treated with DMEM Ø FCS and eLDL (20 µg/mL) and incubated for the indicated time periods. After each incubation period, an MTT assay was performed and the absorbance at 550 nm was measured to determine the viability of the cells. In general, the viability of the cells did not decrease over time. However, slight differences were observed between the different treatments. VICs/myofibroblasts incubated with 20 µg/mL eLDL tended to show a non-significant decrease in viability (Figure 1) compared to treatment with DMEM Ø FCS.

### 3.2. eLDL Upregulates Phosphate Induced Calcification in Cultured VICs/Myofibroblasts

To investigate the role of eLDL in triggering calcification of VICs/myofibroblasts, an alizarin red-based assay was used [34]. Cells were exposed to a calcification medium containing 1 mM inorganic phosphate (Na_2_HPO_4_; PM) wherein no spontaneous calcification was observed. However, when VICs/myofibroblasts cultured in PM were treated with eLDL (2.5 and 5 µg/mL), significantly increased mineralization of eLDL-treated cells was detected as early as day 3 of incubation (Figure 2A).

### 3.3. eLDL and the Expression of Calcification-Related Genes

Next, qPCR analysis was performed to evaluate whether the enhanced calcification in eLDL-treated VICs/myofibroblasts is associated with an alteration of the expression of genes known to be involved in the process of vascular calcification. For this purpose, mRNA was harvested from VICs/myofibroblasts cultured for 7 days in PM with and without eLDL (control). The expression levels of both genes known to promote calcification (alkaline phosphatase (ALPL), bone morphogenic protein 2 (BMP-2), Runt-related transcription factor 2 (RUNX2), Osterix/SP7 and osteopontin (OPN)/SPP1)) and genes known to inhibit calcification (matrix Gla protein (MGP) and ectonucleotide pyrophosphatase/phosphodiesterase-1 (ENPP1)) were investigated and normalized to the housekeeping gene (GAPDH) (Figure 2C). eLDL significantly increased gene expression of ALPL (1.4-fold) and SP7 (5-fold). Interestingly, the mRNA level of RUNX2 was reduced by eLDL. At the same time, the expression of ENPP1, a known inhibitor of calcification [41,42], was markedly increased (1.7-fold). Furthermore, we found no significant differences in the mRNA expression levels of the osteogenic factors BMP-2, MGP, and SPP1 in eLDL-treated VICs/myofibroblasts compared to control cells.

Of note, our data showed a strong induction of ANGPTL4 mRNA in response to eLDL-treatment in VICs/myofibroblasts. Exposure of cells to 5 μg/mL eLDL for 7 days increased ANGPTL4 gene expression approximately 45-fold (Figure 2D) demonstrating that eLDL is very potent in inducing ANGPTL4 mRNA.

### 3.4. eLDL and p38 MAPK in Human VICs/Myofibroblasts

The p38 MAPK is involved in inflammatory signaling in various settings and cell types and has gained attention in the field of both atherosclerosis and CAVD (calcific aortic valve disease) research, especially since these cardiovascular diseases have been recognized as active, inflammation-driven processes [14]. Previous work has shown that incubation of different cell types with either native LDL [43] or eLDL [23,24] resulted in p38 MAPK phosphorylation. Accordingly, VICs/myofibroblasts were incubated with eLDL (10 or 20 µg/mL) for 30 min. Medium alone served as a negative control. Sequential WB analyses of cell lysates were performed to detect both p38 MAPK and phospho-p38 MAPK. Phosphorylation site-specific antibodies were used to investigate MAPK activation in human VICs/myofibroblasts. The WB data showed that treatment with eLDL stimulated p38 MAPK phosphorylation. While the amount of p38 protein remained the same, the amount of phosphorylated p38 increased in a dose-dependent manner. Phospho-p38 MAPK showed induction of the phosphorylated protein by eLDL after 30 min, with the signal becoming strongly visible upon incubation with 10 µg/mL and increasing in intensity after incubation with 20 µg/mL (Figure 3A). Thus, eLDL induces phosphorylation of p38 MAPK in a dose-dependent manner. These findings were corroborated by immunohistochemical staining of sclerotic aortic valves showing phosphorylation of p38 MAPK and its specific downstream substrate heat shock protein 27 (Hsp27). Aortic valve immunostaining for phosphorylated and total protein was performed with antibodies for p38 MAPK, phospho-p38 MAPK, Hsp27, and phospho-Hsp27. Increased phosphorylation of p38 MAPK was accompanied by the upregulation of the phosphorylated Hsp27 protein, a potential marker of p38 MAPK activity (Figure 3D), indicating the presence of an activated p38 signaling pathway in aortic valve sclerosis.

The p38 MAPK family is composed of four p38 MAPK isoforms: p38α/MAPK14, p38β/MAPK11, p38γ/MAPK12, and p38δ/MAPK13. To elucidate the role of p38 MAPK signaling in the process of AS, we examined the mRNA expression pattern of each of the four p38 MAPK isoforms in VICs/myofibroblasts. Expression of the different p38 MAPK isoforms in human VICs/myofibroblasts was investigated by qPCR using isoform-specific primers. Analysis of mRNA levels in cells from 10 different patients (Figure 3C) revealed a consistent expression pattern, with p38α/MAPK14 showing the highest relative expression of the p38 MAPK isoforms followed by both p38β/MAPK11 and p38γ/MAPK12, whereas there was only weak expression of p38δ/MAPK13.

### 3.5. Induction of Both IL-6 and IL-33 by eLDL

Concomitant with the activation of p38 MAPK, an upregulation of the cytokines IL-6 and IL-33 was observed after eLDL treatment. Experiments were performed to investigate the impact of eLDL on the mRNA expression pattern of the pro-inflammatory cytokines IL-6 and IL-33 in VICs/myofibroblasts. Human VICs/myofibroblasts from three to four healthy donors were treated with eLDL (20 µg/mL) and incubated for the indicated time periods (Figure 4). Cells without treatment (DMEM) served as controls. The relative expression levels of IL-6 and IL-33 were measured via qPCR analysis. The data showed that cells treated with eLDL had significantly higher mRNA levels for both cytokines in VICs/myofibroblasts than the untreated control (DMEM). Specifically, cytokine expression increased with time, with IL-6 rising already after 30 min (Figure 4A) and IL-33 after 6 h (Figure 4B).

### 3.6. Colocalization of eLDL with IL-6 and IL-33

25 specimens of aortic lesions fulfilling the criteria of Grades 3 and 4 as defined [21] were examined, and similar findings were made in all cases. With the use of a specific mAb, eLDL was detectable in every lesion examined. Grade 3 and Grade 4 showed a predominant extracellular localization of eLDL, mainly around calcified areas and/or cholesterol crystal deposits (Figure 5B). Likewise, both IL-6 (Figure 5C) and IL-33 (Figure 5D) were detectable in every lesion examined, with close intermingling and overlap of the different antigens within and around calcified areas (colored red by Masson–Goldner, Figure 5A).

### 3.7. IL-6 and IL-33 Activate the p38 MAPK Pathway

To elucidate the role of IL-6 and IL-33 in activating the p38 MAPK signaling pathway, primary VICs/myofibroblasts from AS patients were incubated with recombinant IL-6 (100 ng/mL) and IL-33 (100 ng/mL) for 24 h and 48 h, respectively. Medium alone served as the control. Western blot analyses of cell lysates were utilized to detect the activation of the p38 MAPK pathway following the IL-6 and IL-33 treatments, respectively. Western blot analyses showed that IL-6 and IL-33 stimulation markedly increased the phosphorylation level of p38 MAPK as compared to the untreated control. While the amount of total p38 MAPK protein remained unchanged, there was an increase in the amount of phosphorylated p38 MAPK over time (Figure 6). The signal of phosphorylated protein became clearly visible after 24 h incubation with IL-6 (100 ng/mL) or IL-33 (100 ng/mL) and increased in intensity after incubation for 48 h. These results indicate that both cytokines IL-6 and IL-33 induce activation of the p38 MAPK signaling pathway in VICs/myofibroblasts.

### 3.8. The p38 MAPK Pathway Is Involved in Increased IL-6 Cytokine Expression

Pharmacological inhibitors were used to investigate the impact of the p38 MAPK signaling pathway on the expression of IL-6 and IL-33 in response to eLDL treatment. VICs/myofibroblasts were treated with the p38 MAPK inhibitors skepinone-L and SB203580 for 2 h prior to treatment with 40 μg/mL eLDL for 24 h. The expression of IL-6 and IL-33 was evaluated by RT-PCR as performed in Figure 4. As shown by the RT-PCR data, pre-treatment with skepinone-L (0.1 μM) and SB203580 (20 μM) prevented the stimulation of IL-6 by eLDL (Figure 7A). On the other hand, neither inhibitor inhibited eLDL-induced IL-33 gene expression (Figure 7B).

### 3.9. IL-33 Induces IL-6 Cytokine Expression in Primary VICs/Myofibroblasts

Experiments were performed to investigate the effects of IL-33 on the expression of the inflammatory cytokine IL-6 in VICs/myofibroblasts. Cells were treated for 6 days with and without IL-33 (100 ng/mL) in combination with PM at a concentration of 1 mM inorganic phosphate. The expression of IL-6 was measured using qPCR. After treatment with human recombinant IL-33 and PM containing 1 mM inorganic phosphate, VICs/myofibroblasts isolated from AS patients showed a significant increase (3-fold) in the mRNA expression of IL-6 (Figure 8A), demonstrating marked induction of IL-6 by IL-33.

### 3.10. Effects of IL-6 on Phosphate-Induced Calcification in VICs/Myofibroblasts

Based on the above observations, we wondered whether IL-6 promotes calcification of the aortic valve. For this purpose, human VICs/myofibroblasts were incubated in PM with inorganic phosphate (1 mM) and treated with and without IL-6 (100 ng/mL) for 6 days, respectively. Alizarin red staining was used to assess mineralization. Cells cultured in PM containing solely 1 mM inorganic phosphate showed no spontaneous calcification. However, treatment of VICs/myofibroblasts with IL-6 (100 ng/mL) incubated with PM containing 1 mM phosphate resulted in positive calcium deposition of IL-6-treated cells after 6 days of incubation (Figure 8B). For quantification, the deposited alizarin–Ca^2+^ complexes were extracted, and the OD was measured at 570 nm and normalized to total cellular protein (Figure 8C).

## 4. Discussion

In the present study, we demonstrated that eLDL enhances phosphate-induced calcification and alters osteogenic gene expression in VICs isolated from diseased aortic valves. We have shown that eLDL induces the activation of p38 MAPK, which has been identified as a pro-inflammatory protein in atherosclerosis [24]. Further analysis of the p38 MAPK signaling pathway revealed that p38α/MAPK14 is the most abundantly expressed p38 isoform in isolated human VICs/myofibroblasts. In parallel to p38 MAPK activation, eLDL also leads to upregulation of IL-6 and IL-33 expression. These findings were corroborated by immunohistochemical staining of calcified aortic valves showing colocalization of eLDL with IL-6 and IL-33. Since eLDL treatment resulted in increased IL-6 and IL-33 expression, we utilized specific p38 MAPK pathway inhibitors (skepinone-L and SB203580) to determine whether activation of this pathway is important for eLDL-induced cytokine production. Our data showed that inhibition of p38 MAPK decreases the expression of IL-6 but not IL-33 following eLDL treatment of VICs/myofibroblasts. Furthermore, our results show that both cytokines IL-33 and IL-6 increase the phosphorylation level of p38 MAPK in primary VICs/myofibroblasts. Moreover, we demonstrated for the first time that IL-33 stimulates IL-6 expression in cells isolated from AS patients. IL-6 in turn, like eLDL, is able to promote calcification of VICs/myofibroblasts treated with inorganic phosphate. Thus, one possible mechanism by which eLDL promotes calcification in VICs/myofibroblasts could be IL-33-mediated IL-6 cytokine production through activation of the p38 MAPK signaling pathway. Elucidating the pathological role of eLDL will contribute to a better understanding of calcification and the development of therapeutic interventions to prevent the disease’s progression.

To examine whether eLDL promotes calcification of VICs/myofibroblasts, cells were cultured in calcification medium to increase susceptibility to calcification. A wide spectrum of alizarin red staining, ranging from severe to almost no calcification, has been detected for human VICs/myofibroblast cultured in PM (1 mM NaH_2_PO_4_) with the addition of the pro-inflammatory stimulus eLDL. Our data demonstrate that VICs/myofibroblasts cultured in PM exhibit donor-to-donor variation.

New evidence suggests that osteogenic differentiation of human VICs/myofibroblasts plays a key role in valve calcification. VICs differentiate into activated myofibroblasts and osteoblast-like cells in response to specific mediators [5,44]. It is likely that many signaling networks interact to drive the pathological transformation of VICs in aortic valve sclerosis, although the detailed molecular processes and their timing are not fully understood. We demonstrated significant differences in the transcriptome of human VICs/myofibroblasts when treated with eLDL. Our functional studies suggest a possible role for eLDL in altering gene expression related to the calcification process by activating osteogenic genes or inhibiting calcification inhibitors. Stimulation of VICs/myofibroblasts cultured in PM with eLDL resulted in upregulation of the expression of the osteogenic marker alkaline phosphatase (ALPL) (1.4-fold increase) and SP7/Osterix (5-fold increase). ALP is known to have a calcification-promoting effect by decreasing levels of the mineralization inhibitor inorganic pyrophosphate (PP_i_) [45,46,47,48]. The ability of ALP to regulate pyrophosphate levels in vivo has been demonstrated by the elevated pyrophosphate levels in ALP-deficient humans [49]. A logical conclusion is that ALP promotes calcification by reducing pyrophosphate levels. SP7, on the other hand, is a transcription factor important for bone formation and osteogenic differentiation [50,51]. It has been shown that a reduction in SP7 inhibits osteogenic differentiation and calcification of human VICs [52]. Furthermore, our data showed that the mRNA expression level of RUNX2 was significantly reduced by eLDL, although it is considered a calcification-promoting gene [34,53]. RUNX2 is a transcription factor highly expressed in calcified aortic valves. While RUNX2 is not normally expressed in aortic valves, numerous studies have demonstrated the expression of RUNX2 in calcified human aortic valves and its association with the pathogenesis of AS [54,55,56]. In comparison, the expression of ENPP1, an established regulator of tissue mineralization and inhibitor of calcification [41,42,45], was markedly increased after eLDL treatment. ENPP1 is an ectoenzyme that converts extracellular ATP to adenosine and thereby generates inorganic pyrophosphate [42]. As stated above, pyrophosphate itself is an important inhibitor of calcification [57]. Previous studies have shown that homozygous ENPP1 deficiency in humans leads to loss of enzyme activity and to a severe form of infantile arterial calcification [41,58]. Surprisingly, no reduction in ENPP1 or MGP, another prominent calcification-inhibitor [59,60,61], was observed in our study. Of all the genes examined, only RUNX2 showed a significant reduction in gene expression. The direct molecular mechanisms by which eLDL promotes calcification of VICs/myofibroblasts are at least partially associated with the acquisition of an osteoblastic gene profile, as previously described for smooth muscle cells (SMCs) [34]. 

In agreement with previous findings in human coronary artery SMCs [34], another important result of our study shows that eLDL strongly induces ANGPTL4 mRNA expression (45-fold increase). ANGPTL4 mRNA is induced by various types of lipids, including fatty acids [62,63] and may serve to protect cells from excessive fat uptake [64]. Therefore, we hypothesize that the uptake of eLDL into VICs/myofibroblasts [21] loads the cells with lipids, such as cholesterol and fatty acids, and, thus, upregulates ANGPTL4 mRNA. ANGPTL4 is a potent inhibitor of the triglyceride-hydrolyzing enzyme lipoprotein lipase (LPL) [65] and, thus, presumably prevents lipid toxicity in fat-loaded cells [62,64]. By repressing LPL activity, ANGPTL4 functions as an important regulator of LPL-mediated lipid uptake into cells [66]. Vice versa, ANGPTL4 deficiency increases the uptake of oxLDL into macrophages and enhances lipid-induced stress [67]. Therefore, upregulation of ANGPTL4 mRNA in VICs/myofibroblasts in response to eLDL may represent a mechanism that impairs lipid uptake and, thus, protects cells from excess fat and lipid toxicity.

As an approach to investigate the effect of eLDL on the pro-inflammatory protein p38 MAPK, we analyzed the activation of the p38 MAPK signaling pathway in eLDL-treated VICs/myofibroblasts. The fact that eLDL induces phosphorylation and activation of p38 MAPK in human epithelial cells [23], human coronary artery SMCs [34], human monocytes, and monocyte-derived macrophages [24] as well as in human VICs/myofibroblasts, calls for further investigation of the cellular response. Activation of the p38 MAPK pathway has been demonstrated in macrophages associated with atherosclerotic lesions by immunohistochemistry in patient derived tissues [24] and in animal models [68]. Several in vitro studies have investigated the biological consequences of p38 MAPK activation in atherosclerosis-associated macrophages. First, p38 MAPK was shown to be part of a positive feedback mechanism that drives foam cell formation. OxLDL induces p38 MAPK activation in macrophages, which in turn promotes LDL uptake by PPARγ-mediated upregulation of LDL uptake receptors, such as CD36 [14,43]. The specific p38 inhibitor SB203580 prevents oxLDL-exposed macrophages from becoming foam cells. p38α/MAPK14, the most abundantly expressed isoform in human monocytes [24], is the physiologically relevant isoform of p38 MAPK involved in inflammatory responses [69]. Interestingly, skepinone-L inhibits eLDL-induced activation of p38α MAPK and expression of CD36, with no net effect on foam cell formation [24]. However, the data concerning the role of p38 MAPK in lipoprotein uptake by human monocytes and macrophages are generally sparse and contradictory. Although other studies indicate that LDL uptake by monocytes [70] or cholesterol ester accumulation in macrophages [71] are promoted by p38 MAPK, a further report contradicts this and suggests that inhibition of p38 MAPK has no apparent effect on lipid accumulation in LDL-treated THP-1 cells [72]. In VICs/myofibroblasts, p38 MAPK seems to be involved in important osteogenic signaling pathways and to support the osteogenic differentiation of cells that drive tissue calcification [73,74].

Among the multiple downstream activities of phosphorylated p38 MAPK, cytokine production is a critical component associated with atherosclerosis [75]. In the first instance, we investigated whether eLDL treatment of VICs/myofibroblasts induces cytokines known to be involved in the pathogenesis of AS.

As for the pro-inflammatory cytokines IL-6 and IL-33, there is initial evidence of an association with calcific AS [27,28]. IL-33 is a newly discovered cytokine that belongs to the IL-1 cytokine family [76] and has been demonstrated to play interesting roles in various cardiovascular disease processes, including myocardial infarction, atherosclerosis, and cardiac fibrosis [28,35,77]. On closer examination, however, the data seem to be contradictory, especially when taking a comparative look at atherogenesis. On the one hand, IL-33 is thought to have atheroprotective properties [29,30]. Several processes may be responsible for these properties, including a change in T cell polarization from Th1 to Th2, induction of Th2-cytokines and of protective oxLDL antibodies [29], as well as prevention of macrophage foam cell production [30]. On the other hand, pro-atherogenic actions have also been described in experimental studies [78]. Clinically, serum IL-33 levels are increased in individuals with unstable angina pectoris and acute myocardial infarction compared to stable angina and control groups [31]. Likewise, the effect of the cytokine in the pathogenesis of AS is not yet clear. Many studies have proposed that inflammation-associated factors can promote the unidirectional differentiation of VICs into myofibroblasts or osteoblasts, thereby contributing to valve thickness and calcific nodule formation [28]. In addition, p38 MAPK was recently shown to be involved in IL-33 signaling in macrophages [79]. Our work suggests that IL-33 leads to an upregulation of the pro-inflammatory cytokine IL-6, potentially through stimulation of the p38 MAPK pathway in human VICs/myofibroblasts. Several studies have demonstrated that IL-33 induces the expression of type-1 and -2 cytokines via the p38 MAPK pathway in innate lymphoid cells [80], natural killer cells [81], and leukemia cells [36]. In other cell types, IL-33-mediated expression of the type-2 cytokine IL-6 could be abolished by the pharmacological p38 inhibitor SB203580, suggesting the involvement of p38 MAPK in this process [35,36].

IL-6 is a pro-inflammatory peptide associated with various aspects of cardiovascular disease, but its role in AS in particular is still under debate. We have shown that treatment of cells with both eLDL and IL-33 leads to an increase in IL-6 mRNA. High expression of IL-6 in the aortic valve has been shown to promote mineralization [27]. Furthermore, polymorphisms in the IL-6 gene are associated with AS [82] and also act as a promoter of atherosclerosis [83]. In vitro studies have demonstrated the relationship of IL-6 with BMP-2 and RUNX2, which are considered to be important regulators and promoters of osteogenesis and key elements of the calcification process by controlling the osteogenic transition of VICs in CAVD [27]. It is possible that IL-6, like eLDL, is able to enhance calcification by mediating the osteogenic program in aortic valves.

In summary, our study shows that the expression of IL-6 and IL-33 is elevated in primary VICs/myofibroblasts treated with the lipoprotein modification eLDL. Furthermore, the data support a role for the p38 MAPK pathway in IL-33-induced cytokine release. Based on these observations, we propose that IL-33 is an important regulator of eLDL-mediated calcification by activating p38 MAPK pathways, which in turn promotes increased IL-6 expression. Thus, collectively, our work suggests that targeting the IL-33/p38 MAPK/IL-6 axis has the potential to be an effective treatment for AS. The lack of success so far in treating AS highlights the need for further research into the molecular pathways leading to valve calcification in order to find new therapeutic targets.

## Figures and Tables

**Figure 1 biomolecules-13-01091-f001:**
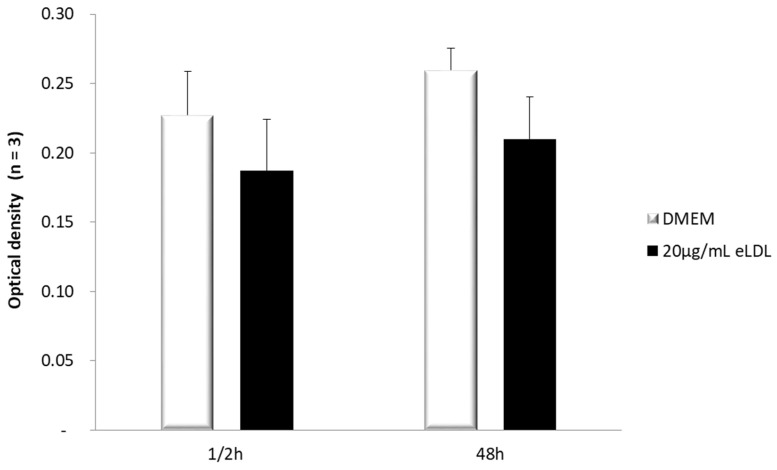
Viability of VICs/myofibroblasts after eLDL treatment (*n* = 3). The VICs/myofibroblasts were seeded on 96-well plates. The cells were treated with DMEM Ø FCS or 20 µg/mL eLDL and incubated for the indicated time periods. After each incubation period, an MTT assay was performed and the absorbance at 550 nm was measured to determine the viability of the cells.

**Figure 2 biomolecules-13-01091-f002:**
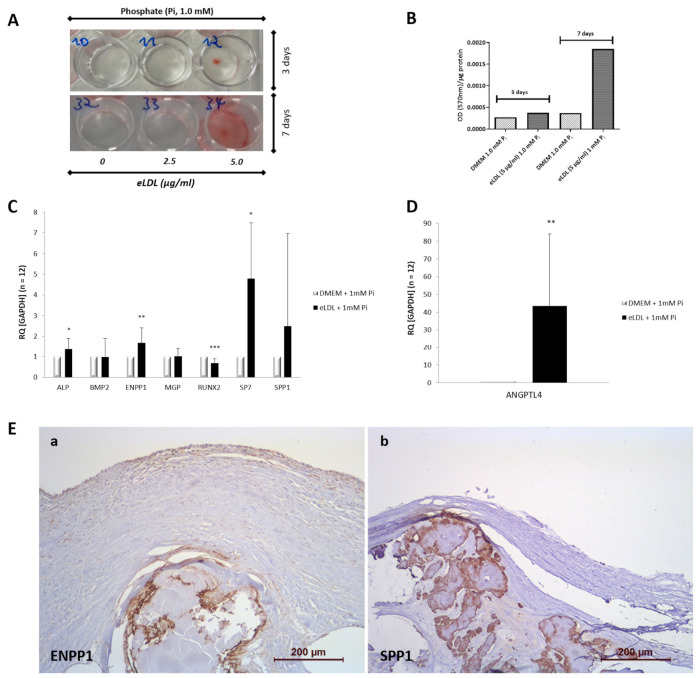
eLDL enhances phosphate-induced calcification in cultured human VICs/myofibroblasts. (**A**) Representative example of confluent monolayers of human VICs/myofibroblasts treated with 1 mM phosphate containing pro-calcifying medium (PM) with (wells 11 & 33 containing 2.5 µg/mL eLDL, wells 12 & 34 containing 5 µg/mL eLDL) or without eLDL (wells 10 & 32) as indicated. Cells cultured in PM containing 1 mM inorganic phosphate (P_i_) served as controls. Cells were fixed and calcium phosphate deposits were stained with alizarin red pH 4.4. (**B**) For quantification of the alizarin red staining, the alizarin–Ca^2+^ complexes were extracted by addition of CPC. The amount of released dye was measured by spectrophotometry at 570 nm. (**C**) qPCR analysis of osteogenic gene mRNA in cultured VICs/myofibroblasts incubated for 7 days in PM containing 1 mM inorganic phosphate (P_i_) in the presence (5 µg/mL) or absence of eLDL (control). The mRNA expression levels were normalized to GAPDH according to the 2^−ddCT^ method. Results from 12 independent experiments are shown. Bar values are means ± SD. (**D**) eLDL induced ANGPTL4 gene expression in VICs/myofibroblasts. ANGPTL4 gene expression was determined for cells exposed to eLDL (5 μg/mL) in combination with PM 1 mM P_i_ for 7 days. Untreated cells (PM with 1 mM P_i_) served as control. Results are presented as means ± SD, *n* = 12, *** *p* < 0.001, ** *p* < 0.01, and * *p* < 0.05. (**E**) Immunohistochemical analysis of osteogenic proteins in AS. Representative sections of Grade 4 aortic valve calcification for (**a**), ENPP1, and (**b**), SPP1. Note the predominant localization of the different antigens around calcified areas. In all panels, the fibrosa with the aortic side of the valve is to the top.

**Figure 3 biomolecules-13-01091-f003:**
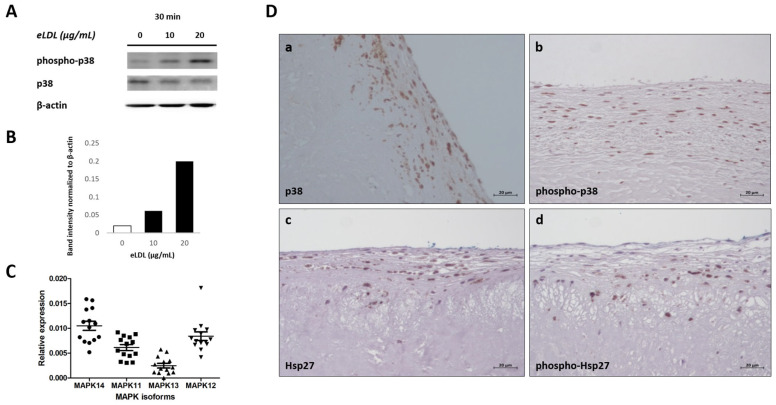
p38 MAPK and eLDL in human aortic valves. (**A**) Detection of rapidly phosphorylated p38 MAPK upon treatment of VICs/myofibroblasts with eLDL. Cells were exposed to 10 or 20 µg/mL eLDL for 30 min. Cells without treatment served as control. Phosphorylated p38 MAPK and total p38 MAPK in whole-cell lysates were sequentially detected by Western blot analysis on the same membrane. Blots were re-probed with β-actin antibody to confirm equal loading. (**B**) WB quantification of the phosphorylation of p38 MAPK. (**C**) Expression of p38 MAPK isoforms in VICs/myofibroblasts. Relative mRNA levels for p38 subtypes quantified in VICs/myofibroblasts from 10 different patients using real-time PCR. Expression levels are given as RQ of target genes (p38α/MAPK14, p38β/MAPK11, p38γ/MAPK12, p38δ/MAPK13) normalized to the reference gene GAPDH. (**D**) Representative immunohistochemical staining of aortic valves sections stained with (**a**), anti-p38; (**b**), anti-phospho-p38; (**c**), anti-Hsp27; (**d**), anti-phospho-Hsp27. The fibrosa with the aortic side of the valve is on top.

**Figure 4 biomolecules-13-01091-f004:**
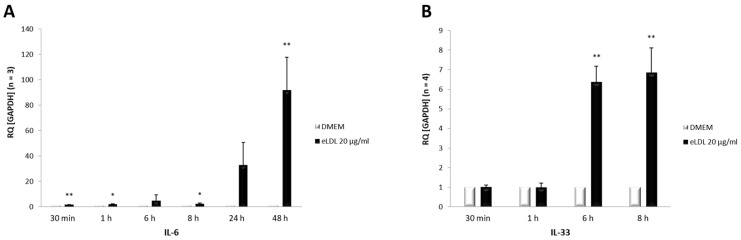
eLDL increases mRNA expression of (**A**) IL-6 and (**B**) IL-33 in cultured human VICs/myofibroblasts. Confluent monolayers of VICs/myofibroblasts were exposed to 20 µg/mL eLDL for the indicated time periods. Cells cultured in DMEM alone were used as controls. Total RNA was extracted, and levels of IL-6 and IL-33 mRNA were measured via qPCR analysis. The expression levels of IL-6 and IL-33 mRNA were normalized to the housekeeping gene (GAPDH). Data are represented as mean ± SD, ** *p* < 0.01, and * *p* < 0.05.

**Figure 5 biomolecules-13-01091-f005:**
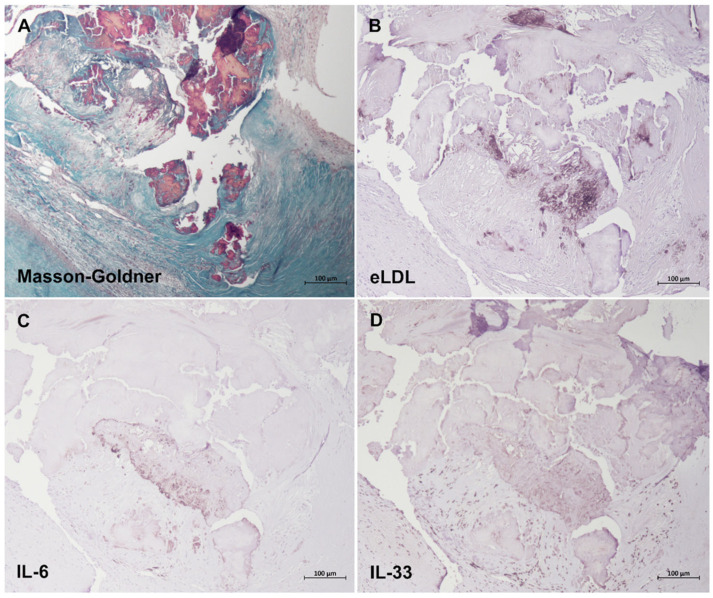
Colocalization of eLDL with IL-6 and IL-33. Representative sequential sections of Grade 4 aortic valve calcification stained with (**A**) Masson–Goldner for (**B**) eLDL, (**C**) IL-6, and (**D**) IL-33. Note the close intermingling and overlap of the different antigens within and around calcified areas (colored red by Masson–Goldner). In all panels, the fibrosa with the aortic side of the valve is to the upper right-hand corner.

**Figure 6 biomolecules-13-01091-f006:**
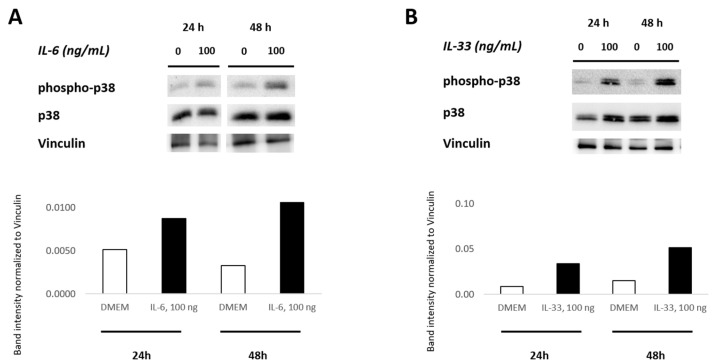
Activation of the p38 MAPK pathway in human VICs/myofibroblasts stimulated with IL-6 and IL-33. Cells from AS patients were treated with (**A**) 100 ng/mL IL-6 and (**B**) 100 ng/mL IL-33 for 24 h and 48 h. Cells without treatment served as controls. Phosphorylated p38 MAPK and total p38 MAPK protein expression were detected by WB analysis. The expression of vinculin was used for loading control. The bar graphs show the relative quantification of phospho-p38 protein compared to the untreated cells.

**Figure 7 biomolecules-13-01091-f007:**
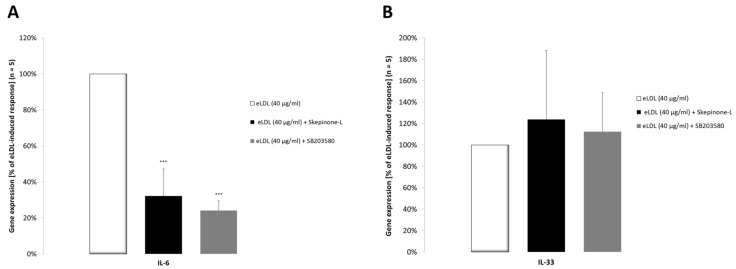
Effect of skepinone-L and SB203580 on the eLDL-induced expression of IL-6 and IL-33 in human VICs/myofibroblasts. Cells were incubated with skepinone-L (0.1 μM) or SB203580 (20 μM) for 2 h and further exposed to 40 μg/mL eLDL for 24 h. mRNA expression levels of (**A**) IL-6 and (**B**) IL-33 were measured via qPCR analysis. The expression levels of IL-6 and IL-33 mRNA were normalized to the housekeeping gene (GAPDH). The eLDL-stimulated cells were assigned an arbitrary value of 100, and the other gene expression values are shown in proportion to this as a percentage. Data are represented as mean ± SD, *** *p* < 0.001.

**Figure 8 biomolecules-13-01091-f008:**
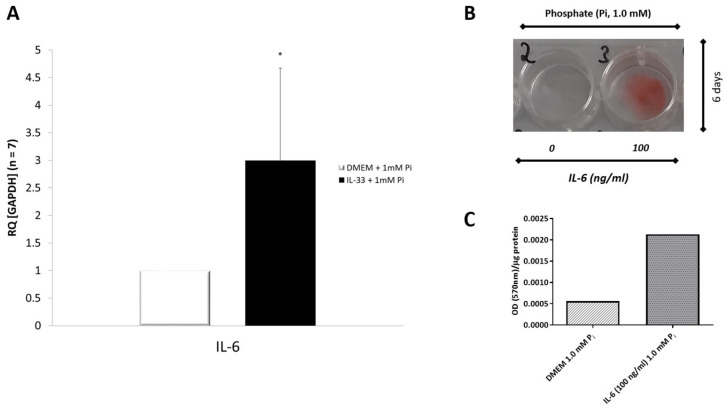
IL-6, IL-33, and calcification. (**A**) IL-33 induces IL-6 mRNA expression in human VICs/myofibroblasts. qPCR analysis of IL-6 mRNA in cultured VICs/myofibroblasts incubated for 6 days in PM containing 1.0 mM inorganic phosphate (P_i_) in the presence (100 ng/mL) or absence of IL-33 (control). The mRNA expression level of IL-6 was normalized to GAPDH according to the 2^−ddCT^ method. Results from 7 independent experiments are shown. The bar values are mean values ± SD, * *p* < 0.05. (**B**) Effects of IL-6 on phosphate-induced calcification in VICs/myofibroblasts. Confluent monolayers of human VICS/myofibroblasts cultured in 1 mM phosphate containing PM were treated with (well 3 containing 100 ng/mL IL-6) and without IL-6 (well 2) for 6 days. Calcium phosphate deposits were visualized by alizarin red staining (pH 4.4). (**C**) To quantify calcium deposition, alizarin–Ca^2+^ complexes were extracted by adding CPC. The optical density of the extract was measured at 570 nm and normalized to cellular protein content.

**Table 1 biomolecules-13-01091-t001:** Relations of clinical variables (*n* = 23).

Characteristic	Value
Age, y	66 (6)
BMI, kg/m^2^	25.15 (5.38)
Cholesterol, mg/dL	186 (65.25)
Creatinine, mg/dL	0.9 (0.3)
CRP, mg/dL	0.2 (0.35)
Hb_A1c_, %Hb	5.8 (0.4)
LDL, mg/dL	79.0 (65.0)
N-terminal of the prohormone brain natriuretic peptide, pg/mL	496.0 (890.5)
Gender	
Male	14 (60.87%)
Female	9 (39.13%)
Risk Factor	
Hypertension	12 (52.17%)
Smoking	6 (26.09%)
Hyperlipidemia	4 (17.39%)
Hypercholesterolemia	5 (21.74%)
Diabetes mellitus	5 (21.74%)

Data are presented as median and interquartile range or number and percentage. BMI indicates body mass index; CRP, C-reactive protein; LDL, low-density lipoprotein.

**Table 2 biomolecules-13-01091-t002:** The primary antibodies used for immunohistochemistry and Western blot analysis.

Antigen	Name	Application	Dilution	Source	Company
p38 MAPK	p38 MAPK (D13E1) mAb	IHC-P WB	1:1000 1:1000	Rabbit	Cell Signaling Technology, Danvers, MA, USA
p-p38 MAPK	Phospho-p38 MAPK (Thr180/Tyr182) (D3F9) mAb	IHC-P WB	1:100 1:1000	Rabbit	Cell Signaling Technology
Hsp27	HSP27 (G31) mAb	IHC-P	1:200	Mouse	Cell Signaling Technology
p-Hsp27	Phosopho-HSP27 (Ser82) (D1H2F6) mAb	IHC-P	1:200	Rabbit	Cell Signaling Technology
IL-6	IL-6 Ab polyclonal	IHC-P	1:100	Rabbit	Affinity Biosciences, Cincinnati, OH, USA
IL-33	IL-33 monoclonal antibody (12B3C4)	IHC-P	1:100	Mouse	Invitrogen by Thermo Fisher Scientific, Waltham, MA, USA
eLDL	LDL-8 (AIL-3) mAb	IHC-P	1:2000	Mouse	Torzewski et al. [19]
Vinculin	Vinculin monoclonal antibody (2B5A7)	WB	1:1000	Mouse	Proteintech, Planegg-Martinsried, Germany
β-actin	Monoclonal anti-β-actin clone AC-15	WB	1:1000	Mouse	Sigma-Aldrich, St. Louis, MO, USA
ENPP1	Ectonucleotide pyrophosphatase/phosphodiesterase 1 (ENPP1) antibody	IHC-P	1:25	Rabbit	Antikörper-online (abbexa), Aachen, Germany
SPP1	Osteopontin (SPP1) antibody	IHC-P	1:500	Mouse	Antikörper-online (abbexa)

**Table 3 biomolecules-13-01091-t003:** List of TaqMan gene expression assays.

Protein	Gene	Assay ID
p38α MAPK	*MAPK14*	Hs01051152_m1
p38β MAPK	*MAPK11*	Hs00177101_m1
p38δ MAPK	*MAPK13*	Hs00234085_m1
p38γ MAPK	*MAPK12*	Hs00268060_m1
GAPDH	*GAPDH*	Hs02786624_g1
RUNX2	*RUNX2*	HS01047973_m1
BMP-2	*BMP-2*	Hs00154192_m1
SP7	*SP7*	Hs00541729_m1
ENPP1	*ENPP1*	Hs01054040_m1
MGP	*MGP*	Hs00969490_m1
SPP1	*SPP1*	Hs00959010_m1
ALPL	*ALPL*	Hs01029144_m1
ANGPTL4	*ANGPTL4*	Hs01101127_m1

## Data Availability

The data presented in this study are available upon reasonable request from the corresponding author.

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
