# Peer review of "Pro-Calcifying Role of Enzymatically Modified LDL (eLDL) in Aortic Valve Sclerosis via Induction of IL-6 and IL-33"

_biomolecules, 2023, doi:10.3390/biom13071091_

Round 1
Reviewer 1 Report
The article gives the impression of a well-thought-out and carefully executed work. However, I have questions about choosing an LDL modification. From the introduction it is not clear what is ldl. Is any evidence of present such LDL in vivo? Why did you not use oxLDL? The eLDL was described in your previous work. However, this type of LDL is not common in researchers. Thus, it will be valuable if you add this information to your introduction part.
This information would help readers to understand your study.
Author Response
Comments and Suggestions for Authors
The article gives the impression of a well-thought-out and carefully executed work. However, I have questions about choosing an LDL modification. From the introduction it is not clear what is ldl. Is any evidence of present such LDL in vivo? Why did you not use oxLDL? The eLDL was described in your previous work. However, this type of LDL is not common in researchers. Thus, it will be valuable if you add this information to your introduction part.
This information would help readers to understand your study.
We thank for this reasonable comment and have now provided additional information about eLDL in the introduction part (page 2, line 56 to line 65).

Reviewer 2 Report
The authors were committed to elucidating the pro-calcifying role of eLDL in aortic valve sclerosis. However, there are several major issues in this study that have raised my concerns.
1. On Page 3 Line 130, “Cells were cultured in the absence or presence of inorganic phosphate (1 mM), treated with eLDL (5 μg/ml), IL-6 (10 ng/ml and 100 ng/ml) or IL-33 (10 ng/ml and 100 ng/ml) and harvested after the indicated time points.” The concentration of interventions is a matter worth discussing. Unfortunately, the interventions used in this study were inconsistent and the authors did not provide a detailed explanation. For example, VIC/myofibroblasts were treated with 20 μg/ml eLDL to detect cell viability, but VIC/myofibroblasts were treated with 2.5 or 5.0 μg/ml eLDL to investigate the role of eLDL in calcification.
2. The in-vivo experiments should be done to further confirm the role of eLDL for osteogenic differentiation/calcification of VICs and AS.
3. All pictorial evidence should be quantified as much as possible, including WB, immunohistochemistry, etc.
4. Regarding calcification-related indicators, this study only verified their changes at the RNA level, what about the protein level?
5. There are still many questions worth exploring about the mechanism of eLDL in calcification. Why did the authors choose IL-6, and IL-33, and not others? Current evidence is insufficient to elucidate the upstream and downstream relationships of the IL-33/p38 MAPK/IL-6 axis.
6. Their data showed strong induction of ANGPTL4 mRNA in response to eLDL-treatment in VICs/myofibroblasts. The authors could further explore its related mechanism, and whether eLDL regulates VIC calcification through ANGPTL4.
7. Basic clinical information about the patients and specimens should be provided, and the sample size of each group and corresponding statistics should be labeled.
Author Response
Comments and Suggestions for Authors
The authors were committed to elucidating the pro-calcifying role of eLDL in aortic valve sclerosis. However, there are several major issues in this study that have raised my concerns.
- On Page 3 Line 130, “Cells were cultured in the absence or presence of inorganic phosphate (1 mM), treated with eLDL (5 μg/ml), IL-6 (10 ng/ml and 100 ng/ml) or IL-33 (10 ng/ml and 100 ng/ml) and harvested after the indicated time points.” The concentration of interventions is a matter worth discussing. Unfortunately, the interventions used in this study were inconsistent and the authors did not provide a detailed explanation. For example, VIC/myofibroblasts were treated with 20 μg/ml eLDL to detect cell viability, but VIC/myofibroblasts were treated with 2.5 or 5.0 μg/ml eLDL to investigate the role of eLDL in calcification.
The concentrations of the different interventions refer to amounts already used in previous publications. Furthermore, in this study, the concentration of each intervention was adjusted to the different experiments. For example, for the MTT assay, the VIC/myofibroblasts were treated with 20 μg/ml eLDL to show that this eLDL concentration does not affect cell viability. The following RT-PCR and WB experiments were performed with this "maximum" eLDL concentration (20 μg/ml) determined by the MTT assay. To investigate the role of eLDL in calcification, VICs/myofibroblasts were incubated in combination with inorganic phosphate for an extended period of 6-7 days (calcification assay). Here, the concentration of eLDL (2.5 or 5.0 μg/ml) had to be adjusted (cf. Chellan 2018) to ensure the survival of the cells until the end of the experiment.
- The in-vivo experiments should be done to further confirm the role of eLDL for osteogenic differentiation/calcification of VICs and AS.
Indeed, conducting in-vivo and further ex-vivo and in-vitro experiments would be beneficial to further confirm the role of eLDL in the calcification of VICs. Immunohistological studies, as well as WB analyses of the individual osteogenic genes (see RT-PCR Figure 2) are already in progress. However, completion within the given time frame will unfortunately not be possible. Therefore, these data will be used for further publications.
.
- All pictorial evidence should be quantified as much as possible, including WB, immunohistochemistry, etc.
The quantification for the WB in Figure 3 A was inserted (Figure 3 B) and supplemented accordingly in the figure description.
- Regarding calcification-related indicators, this study only verified their changes at the RNA level, what about the protein level?
In this manuscript, we were able to initially show that eLDL in VICs/myofibroblasts is able to alter the gene expression of known calcification-related genes. Although an investigation at the protein level is also very interesting, we focused mainly on ex-vivo experiments (immunohistochemistry) in addition to mRNA expression. In further experiments, verification at the protein level is planned.
- There are still many questions worth exploring about the mechanism of eLDL in calcification. Why did the authors choose IL-6, and IL-33, and not others? Current evidence is insufficient to elucidate the upstream and downstream relationships of the IL-33/p38 MAPK/IL-6 axis.
Indeed, many questions about the mechanism of eLDL in calcification remain unexplored. In addition to eLDL, we chose to investigate the two pro-inflammatory cytokines IL-6 and IL-33 in this manuscript, as there is already preliminary evidence for both cytokines to be associated with calcified aortic valve stenosis (El Husseini 2014, He, Guo 2020). Further experiments to clarify the upstream and downstream relationships of the IL-33/p38 MAPK/IL-6 axis were added to the manuscript (page 12, line 403 to page 13, line 419). An important point here was to clearly demonstrate the involvement of p38 MAPK in eLDL-induced IL-6 and IL-33 expression as well as calcification. This was achieved by using appropriate pharmacological p38 MAPK inhibitors (skepinone-L and SB203580) (see Figure 7).
- Their data showed strong induction of ANGPTL4 mRNA in response to eLDL-treatment in VICs/myofibroblasts. The authors could further explore its related mechanism, and whether eLDL regulates VIC calcification through ANGPTL4.
This is a very good and important point. Due to the strong induction of ANGPTL4 by eLDL, it is of great interest to study the effects of ANGPTL4 expression and the mechanisms involved. We are currently investigating the relationship between eLDL, ANGPTL4 and calcification. We are interested in a potential molecular mechanism in eLDL-induced upregulation of ANGPTL4 mRNA in VICs/myofibroblasts. Among other things, we want to test whether lipid loading of VICs/myofibroblasts with eLDL (foam cell formation) is required for up-regulation of ANGPTL4 mRNA. Furthermore, immunohistochemical detection will also be performed. The results are then to be used for later publications, as the implementation is very time-consuming and the scope would go beyond the scope of this manuscript.
- Basic clinical information about the patients and specimens should be provided, and the sample size of each group and corresponding statistics should be labeled.
Thank you for your comment. A table with the basic clinical data of the patients and the samples used has been added to the manuscript (see Table 1; page 2, line 93 to line 95). The information about the samples size is in the graph or in the description of the graph.
Reviewer 3 Report
The study by Witz and colleagues delineates molecular mechanisms underlying the pathogenesis of aortic valve stenosis (AS). The study design elegantly combines the in vitro and ex vivo approach and the experimental methodology is sound. The results of the study are interesting and provide novel insights into the pathogenesis of AS. The manuscript is generally well-written and illustrational material is of high quality. There are, however, few points, which should be addressed be the authors to further improve the manuscript:
1.) The major point is to provide a clear-cut evidence for the involvement of p38 MAPK in the eLDL-induced IL-33 and IL-6 production as well as calcification. This can be achieved by application of appropriate synthetic kinase inhibitors with known pharmacological properties.
2.) By the same token, the authors may consider to discriminate between the relative contribution of eLDL, IL-33 and IL-6 to the calcification process by downregulating IL-6 and IL-33-induced signalling using inhibitors or supressing the expression of the respective receptors with siRNA.
3.) The Discussion section of the manuscript is quite lengthy and might be condensed by approx. 30 %. Similarly, the reference list would easily satisfy the requirements of a review manuscript, but for the original contribution the authors could limit it to the most pertinent positions.
Author Response
Comments and Suggestions for Authors
The study by Witz and colleagues delineates molecular mechanisms underlying the pathogenesis of aortic valve stenosis (AS). The study design elegantly combines the in vitro and ex vivo approach and the experimental methodology is sound. The results of the study are interesting and provide novel insights into the pathogenesis of AS. The manuscript is generally well-written and illustrational material is of high quality. There are, however, few points, which should be addressed be the authors to further improve the manuscript:
1.) The major point is to provide a clear-cut evidence for the involvement of p38 MAPK in the eLDL-induced IL-33 and IL-6 production as well as calcification. This can be achieved by application of appropriate synthetic kinase inhibitors with known pharmacological properties.
Indeed, clear evidence for the involvement of p38 MAPK in eLDL-induced IL-33 and IL-6 production was previously lacking in this manuscript. An important experiment has been added to the manuscript demonstrating the involvement of p38 MAPK in eLDL-induced IL-6 and IL-33 expression (page 12, line 403 to page 13, line 419). To investigate the impact of the p38 MAPK pathway on IL-6 and IL-33 expression, VICs/myofibroblasts were treated with the two p38 MAPK inhibitors skepinone-L and SB203580 (see Figure 7).
2.) By the same token, the authors may consider to discriminate between the relative contribution of eLDL, IL-33 and IL-6 to the calcification process by downregulating IL-6 and IL-33-induced signalling using inhibitors or supressing the expression of the respective receptors with siRNA.
The downregulation of IL-6 and IL-33-induced signalling using inhibitors or siRNA is a very good idea that we will certainly keep in mind for subsequent experiments. As for the present manuscript, we refer to our additional experiments with the two p38 MAPK inhibitors (see above).
3.) The Discussion section of the manuscript is quite lengthy and might be condensed by approx. 30 %. Similarly, the reference list would easily satisfy the requirements of a review manuscript, but for the original contribution the authors could limit it to the most pertinent positions.
Thank you very much for this comment. We have shortened the discussion section and adjusted the reference list accordingly.
Round 2
Reviewer 2 Report
-
The manuscript has been partially improved. However, the issue of eLDL, IL-6, or IL-33 concentration in the current study remains unsolved. In addition, the verification at protein level need to be done.
Author Response
We have now included adjustments and an explanation of the different concentrations in the Materials & Methods section (page 5 line 146 to line 157 and page 6 line 195 to line 198). The various doses of eLDL and the dose of the two cytokines IL-6 and IL-33 used in this study were determined based on preliminary experiments and previously published in vitro experiments (cf. Chellan et al., 2018 (2.5 and 5.0 μg/ml eLDL), Zhu & Carver, 2012 and Wang et al., 2020 (100ng/ml IL-33)). The concentration of IL-6 and IL-33 was standardised here (IL-6 100 ng/ml and IL-33 100 ng/ml). In comparison, the concentration of eLDL was adjusted to the different experiments (MTT assay, RT-PCR and WB (20 μg/ml); calcification assay (2.5 or 5.0 μg/ml) for reasons of ensuring cell survival and to better clarify the results.
With regard to the indicators related to calcification, we have further investigated, the changes at the protein level in addition to the RNA level. In particular, we have now performed additional immunohistochemical experiments for ENPP1 and SPP1 (see figure 2 E).
Reviewer 3 Report
In the revised manuscript the authors most satisfactorily responded to my original criticism. My last suggestion for them is to improve the graphical presentation of results by enhancing the legibility of figures and unifying fonts and font size across figures and tables.
Author Response
Thank you for the very good point. We have made some changes to the graphical presentation of the results. In order to improve legibility, the font size (labelling, legend and axis scaling) was adjusted, especially with focus on figures 3, 7 and 8.